# Primary Hyperparathyroidism in Homozygous Sickle Cell Patients: A Hemolysis-Mediated Hypocalciuric Hypercalcemia Phenotype?

**DOI:** 10.3390/jcm10215179

**Published:** 2021-11-05

**Authors:** Edmat Akhtar Khan, Lynda Cheddani, Camille Saint-Jacques, Rosa Vargas-Poussou, Vincent Frochot, Remi Chieze, Emmanuel Letavernier, Virginie Avellino, Francois Lionnet, Jean-Philippe Haymann

**Affiliations:** 1Service de Néphrologie, Université de Lorraine, CHRU-Nancy, 54500 Vandœuvre-lès-Nancy, France; edmat.khan@gmail.com; 2Unité HTA, Prévention et Thérapeutique Cardiovasculaires, Assistance Publique—Hôpitaux de Paris, Hôpital Hôtel Dieu, 75004 Paris, France; lynda.cheddani@aphp.fr; 3Centre de Diagnostic et de Thérapeutique, Hôtel-Dieu, Université de Paris, 75006 Paris, France; 4Service des Explorations Fonctionnelles Multidisciplinaires, Assistance Publique—Hôpitaux de Paris, Hôpital Tenon, 75020 Paris, France; camille.saint-jacques@aphp.fr (C.S.-J.); vincent.frochot@aphp.fr (V.F.); remi.chieze@aphp.fr (R.C.); emmanuel.letavernier@aphp.fr (E.L.); 5Unité Mixte de Recherche (UMR) S 1155, Institut National de la Santé et de la Recherche Médicale, Sorbonne Université, Hôpital Tenon, 75020 Paris, France; 6Centre d’Investigation Clinique, Centre de Référence des Maladies Rénales Héréditaires de l’Enfant et de l’Adulte, Assistance Publique—Hôpitaux de Paris, Hôpital Européen Georges Pompidou, 75015 Paris, France; rosa.vargas@aphp.fr; 7Service de Médecine Interne, Centre de Référence de la Drépanocytose, Assistance Publique—Hôpitaux de Paris, Hôpital Tenon, 75020 Paris, France; virginie.avellino@aphp.fr (V.A.); francois.lionnet@aphp.fr (F.L.)

**Keywords:** primary hyperparathyroidism, sickle cell disease, urinary calcium, familial hypocalciuric hypercalcemia, FeCa^2+^, hemolysis

## Abstract

Primary hyperparathyroidism (pHPT) has been reported to have a higher prevalence in sickle cell disease (SCD) patients, including a high rate of recurrence following surgery. However, most patients are asymptomatic at the time of diagnosis, with surprisingly infrequent hypercalciuria, raising the issue of renal calcium handling in SCD patients. We conducted a retrospective study including (1) 64 hypercalcemic pHPT non-SCD patients; (2) 177 SCD patients, divided into two groups of 12 hypercalcemic pHPT and 165 non-pHPT; (3) eight patients with a diagnosis of familial hypocalciuric hypercalcemia (FHH). Demographic and biological parameters at the time of diagnosis were collected and compared between the different groups. Determinants of fasting fractional excretion of calcium (FeCa^2+^) were also analyzed in non-pHPT SCD patients. Compared to non-SCD pHPT patients, our data show a similar ionized calcium and PTH concentration, with a lower plasmatic calcitriol concentration and a lower daily urinary calcium excretion in pHPT SCD patients (*p* < 0.0001 in both cases). Fasting FeCa^2+^ is also surprisingly low in pHPT SCD patients, and thus inadequate to be considered hypercalcemia, recalling the FHH phenotype. FeCa^2+^ is also low in the non-pHPT SCD control group, and negatively associated with PTH and hemolytic biomarkers such as LDH and low hemoglobin. Our data suggest that the pHPT biochemical phenotype in SCD patients resembles the FHH phenotype, and the fasting FeCa^2+^ association with chronic hemolysis biomarkers strengthens the view of a potential pharmacological link between hemolytic by-products and calcium reabsorption, potentially through a decreased calcium-sensing receptor (CaSR) activity.

## 1. Introduction

Homozygous sickle cell disease (SCD), also known as sickle cell anemia, is one of the most prevalent monogenic disorders in the world and is most particularly found in sub-Saharan Africans or people with African ascendance [1]. A single change in aminoacid in the β-globin chain (replacement of a glutamic acid for a valine) is the cause for the formation of hemoglobin S [2], a pathologic variant of normal adult hemoglobin responsible for abnormally shaped erythrocytes, which ultimately leads to a wide array of acute and chronic organ lesions, due to chronic hemolysis and acute vaso-occlusive crisis [3]. Numerous metabolic disorders have also been identified, such as hyperphosphatemia [4,5,6,7,8], hypomagnesemia [9,10] or metabolic acidosis [11,12]. Vitamin D deficiency is a frequent finding among endocrine disorders [13,14], with reported occurrence of secondary hyperparathyroidism [15] and cases of bone fragility [16,17].

Primary hyperparathyroidism (pHPT) is an endocrine disorder caused by the excessive secretion of parathyroid hormone (PTH) by the parathyroid glands, with the diagnosis generally being established on the association of hypercalcemia, elevated PTH levels and high urinary calcium excretion [18]. This disease was not a focus in SCD patients, despite several case reports [19,20], until a recent paper by Denoix et al. reported a 5% prevalence of pHPT [21,22]. SCD patients with pHPT were further analyzed and reported to mainly experience adenoma (90% of patients who underwent surgical treatment) and to be mostly clinically asymptomatic with mild hypercalcemia. On a small series of 12 patients, only three had hypercalciuria.

After metabolic evaluation of several SCD patients in our department with a final diagnosis of pHPT, we were surprised to find a lack of hypercalciuria. This raised the issue of the biochemical features of these patients compared to pHPT renal stone patients and, more broadly, of the determinants of the handling of renal tubular calcium in an SCD population.

## 2. Materials and Methods

This retrospective monocentric study in Tenon Hospital between January 2011 and December 2020 (Assistance Publique des Hôpitaux de Paris, France) included three different cohorts of patients: (1) Renal stone patients referred to our Department of Physiology for hypercalcemia with a final diagnosis of pHPT; (2) Homozygous SCD patients referred to our department for a routine metabolic evaluation; (3) Patients referred for hypercalcemia with a final diagnosis of inactive *CaSR* mutation, also called familial hypocalciuric hypercalcemia (FHH), after gene sequencing. All patients included in this study signed written informed consents. Clinical and biological data were collected from the department of physiology database (data collection was approved by the “Commission Nationale de l’Informatique et des Libertés” according to French legislation, n°2065902v0).

In each group, we collected clinical data such as age, gender, height, weight, blood pressure and current medication. Exclusion criteria were patients with an age < 18 years old, estimated glomerular filtration rate (eGFR) < 60 mL/min/1.73 m^2^ and treatment with thiazide diuretics, bisphosphonates or lithium as these medications may interfere with plasma or urine calcium concentrations. pHPT in both SCD and non-SCD groups was defined by the association of hypercalcemia (serum-ionized calcium ≥ 1.30 mmol/L) and inappropriate intact PTH levels (≥40 pg/mL). A *CaSR* mutation was ruled out by a negative genetic screening in 11 out of 12 patients, and a positive ^99m^Tc sestamibi fixation was found in the remaining patient. Furthermore, positive imaging was detected in 5 out of 5 tested patients. Among the four patients who underwent parathyroidectomy, three patients had normalized calcium and PTH levels, and one patient had secondary hyperparathyroidism (low serum ionized calcium level with an increased serum PTH). Histology was in accordance with a single adenoma in three patients and two adenomas in one patient.

Among 192 eligible SCD patients, 15 were excluded for chronic renal failure. Thus, a total of 177 patients were included and divided into pHPT SCD (*n* = 12) and non-pHPT SCD groups (*n* = 165).

Among 77 eligible renal stone pHPT non-SCD patients, one was excluded for chronic renal failure, and 12 were excluded for either bisphosphonates or thiazide diuretics intake. A total of 64 pHPT non-SCD patients were thus included.

### 2.1. Assays

Biological data were collected at the time of the diagnosis of pHPT. Serum and urine creatinine levels were measured by the enzymatic method on a Kenolab 20 analyzer from Thermo Fisher Scientific. We used a conventional colorimetric assay to measure phosphate, protein, urea, uric acid and glucose on a Kenolab 20 analyzer. Bicarbonate, sodium, potassium and ionized calcium were measured with an ABL 815 from Radiometer. Plasma and urine calcium and magnesium concentrations were measured with the PerkinElmer 3300 atomic absorption spectrometer. 25-hydroxyvitamin D and 1,25-dihydroxyvitamin D levels were measured with radioimmunoassay kits from Immunodiagnostics Systems Ltd. Parathyroid hormone was measured with the ELSA-PTH radioimmunoassay kit from Cisbio International. We measured FGF23 using the C-terminal assay from Immunotopics. Bone turnover was assessed by serum bone alkaline phosphatase level (BAPL) measured with the Ostase bone alkaline phosphatase Enzyme immunoassay from Immunodiagnostics Systems Ltd., and deoxypyridinoline (DPD) was measured by the radioimmunoassay method from Immunodiagnostics Systems Ltd. Urine ammonium (NH_4_^+^) was measured with the RANDOX Laboratories Kit. Other laboratory values were measured using standard hospital laboratory techniques. eGFR was calculated using the CKD-EPI equation [23].

### 2.2. Statistical Analysis

Variables were expressed as percentages, means ± SD, or medians (IQR: interquartile range), as appropriate. Comparisons of quantitative variables between the groups were performed using Kruskal–Wallis test and post-hoc Dunn’s multiple comparison test. In non-pHPT SCD patients, associations between fasting FeCa^2+^ and variables of interest were tested separately using a univariate regression test and a multilinear regression analysis when the adjustment on serum PTH was performed. Statistical testing was conducted at the two-tailed α-level of 0.05. All statistical analyses were carried out using GraphPad Prism 8.4.3 and Statview 5.0 software.

## 3. Results

A diagnosis of pHPT was performed in 6.8% of the 177 SCD patients included in our study. Among the 12 patients with pHPT, we found a median total calcemia of 2.61 mmol/L (IQR: 2.49–2.66); none of the patients had a total calcemia above 2.75 mmol/L.

### 3.1. pHPT SCD Patients versus Non-SCD pHPT Patients

A comparison between pHPT renal stone non-SCD patients and pHPT SCD patients (Table 1) showed no difference for serum calcium (total and ionized) or PTH levels, conversely to 1,25-dihydroxyvitamin D which is lower (*p* < 0.0001) and 24 h calciuria, which is dramatically decreased in pHPT SCD patients (8.7 mmol/day versus 2.3 mmol/day, respectively, *p* < 0.0001). Noticeably, daily sodium chloride intake assessed by 24-h urine sodium is similar, whereas diet protein intake assessed by 24-h urine urea is significantly decreased in pHPT SCD patients. As expected, eGFR is higher in SCD patients (*p* < 0.0001 for both pHPT and non-pHPT SCD). Remarkably, serum phosphate levels in pHPT SCD patients are significantly higher than those of non-SCD pHPT (1.04 mmol/L vs. 0.76 mmol/L, *p* < 0.0001) and significantly lower than those of non-pHPT SCD patients (1.04 mmol/L vs. 1.21 mmol/L, *p* < 0.0001). Moreover, 75% of them have serum phosphate values within a normal range, despite their very high value of circulating FGF23 (median value of 802 RUI/mL, IQR: 589-1355). Bone resorption biomarker deoxypyridinoline is low for the pHPT SCD group, but BALP, a marker of bone calcium efflux, is not significantly different. Notably, fasting FeCa^2+^, a marker of calcium renal handling, is surprisingly low in the setting of hypercalcemia and significantly decreased in the pHPT SCD group (0.88% vs. 2.84%, *p* < 0.0001).

### 3.2. pHPT SCD Patients versus FHH Patients

Eight patients (2 males and 6 females), with a final diagnosis of FHH confirmed by genetic sequencing, are reported in the present study, with a median age of 41 years old (IQR: 31–47). Median ionized calcium and PTH were 1.38 mmol/L (IQR: 1.36–1.42) and 51 pg/mL (IQR: 33–64), respectively. As shown in Figure 1, pHPT SCD patients’ biochemical pattern appears very similar to FHH patients, particularly 24 h urine calcium (median value at 3.42 mmol/day, IQR: 2.21–5.15) and fasting FeCa^2+^ (median value at 1.05%, IQR: 0.52–1.68).

### 3.3. pHPT SCD Patients versus Non-pHPT SCD Patients

Among our whole SCD population, patients with pHPT are older (42.5 vs. 30 years of age), with a similar sex ratio (58% vs. 55% of females). As expected, calcium (both total and ionized) and serum PTH levels are lower in non-pHPT SCD patients. However, surprisingly, FGF23, 1,25-dihydroxyvitamin D, 24-h urinary calcium levels and fasting FeCa^2+^ are not significantly different from pHPT SCD patients.

### 3.4. Non-pHPT SCD Patients: Biochemical Factors Associated with Fasting FeCa^2+^

Among non-pHPT SCD patients, median ionized calcium is 1.21 mmol/L, with only eight patients having hypocalcemia (ionized calcium < 1.15 mmol/L) and a strikingly low fasting FeCa^2+^ of 0.34%. As shown on Table 2, fasting FeCa^2+^ is negatively associated with PTH, combined with several hemolytic biomarkers such as LDH, low hemoglobin and low hematocrit. Noticeably, these associations remain significant after adjustment on PTH with a positive association with aging. No association is found with eGFR or 1,25-dihydroxyvitamin D.

## 4. Discussion

Our data show a high prevalence of pHPT, reaching 6.8% of our SCD patients, consistent with previous reports [21,22]. To our knowledge, our study is the first to point out that pHPT SCD patients have a similar biological phenotype to that of FHH patients, which is characterized by hypercalcemia with inappropriate PTH levels and low urinary calcium excretion [24,25]. A low FeCa^2+^ is a striking feature, raising the issue of a potential *CaSR* inactivating mutations in our hypercalcemic pHPT SCD patients, which was reasonably ruled out in eleven cases by genetic sequencing, and by the positive imaging of one hyperfunctional parathyroid gland in the remaining patient. This finding is significant, as various guidelines recommend urinary calcium excretion measurement to make the final diagnosis between pHPT and FHH, and consider hypercalciuria as one criterion among others to determine the need for surgery in asymptomatic forms of pHPT [26,27,28,29,30]. More specific guidelines for the SCD population should thus be reconsidered in the future.

The reasons 24 h calciuria was significantly higher in pHPT renal stone patients compared to pHPT SCD patients may be related to a higher calcium intake [31], and/or a higher protein intake [32], as assessed by 24 h urea and NH_4_^+^ excretion (Table 1). Although no diet survey for calcium intake was available, we could assume that intestinal calcium absorption is increased in pHPT renal stone patients, as calcitriol was significantly increased. Nevertheless, in SCD patients, decreased calcium intake and intestinal absorption leading to low urinary calcium may also be due to ethnic origins, as previously reported [33,34,35].

However, a lower fasting FeCa^2+^ in pHPT SCD patients compared to pHPT renal stone patients, despite a higher calcium filtered load (i.e., similar ionized calcium but higher eGFR), suggests a higher tubular calcium reabsorption. A similar PTH concentration and a lower calcitriol in the pHPT SCD group can reasonably rule out an increased calcium reabsorption in the distal tubules [36]. Nevertheless, an increased soluble klotho concentration, which has yet to be confirmed, could account for the increase in TRPV5 activity in this tubular segment, independent of PTH and calcitriol concentrations [37,38]. Alternatively, enhanced TRPV5 expression could also be due to FGF23 activity [39]. Conversely, the thick ascending loop of Henle (TALH) and CaSR could potentially be key components. Indeed, this was suggested by the FHH-like biological phenotype of the pHPT SCD patients, as mentioned above. This hypothesis of an increased tubular reabsorption in calcium and a lower CaSR activity was already suggested in pHPT non-SCD African American patients [40], thus raising the issue of the mechanism at play.

Our data show that, in non-pHPT SCD patients, a negative association was found between fasting FeCa^2+^ and several biomarkers of hemolysis such as LDH, low hemoglobin or low hematocrit patients after adjustment on PTH, and thus independent of PTH-mediated distal tubular calcium reabsorption. It appears that chronic hemolysis, which is strongly associated with hyperfiltration and albuminuria in this population [41,42], would also be associated with an increased calcium tubular reabsorption. Although chronic hemolysis in SCD patients is associated with a systemic macro- and microangiopathy, which leads to organ damages such as pulmonary hypertension, leg ulcers or priapism [43,44], no link was found with renal function or albuminuria, which have been reported to be associated with chronic hemolysis [41,42]. Moreover, renal medulla and TALH lesions, which are considered a consequence of vaso-occlusive events, are an unlikely hypothesis to explain the partial loss of CaSR activity, resulting in enhanced tubular calcium reabsorption. Indeed, no association was found between fasting FeCa^2+^ and fasting urine osmolality or urinary ammonium excretion (data not shown). Besides, an efficient paracellular calcium reabsorption in this tubular segment requires intact claudins 16 and 19 [45]. In accordance with this view, aging is independently associated with a decreased tubular reabsorption, with no link found between fasting FeCa^2+^ and eGFR decrease. Obviously, the identification of the calcium renal tubular segment(s) that account for the low FeCa^2+^ and the hemolytic by-product(s) involved in this process have yet to be unraveled.

Though a lower calcitriol in the pHPT SCD group compared to pHPT renal stone group does not explain a lower fasting FeCa^2+^, it may account for a lower 24 h calcium excretion through a decrease in calcium intestinal absorption, as mentioned above. Indeed, calcitriol levels are increased only in one-third of pHPT SCD patients but in most pHPT renal stone patients, as expected [46,47,48,49]. This subnormal calcitriol value may be surprising at first glance, as no difference is detected for PTH values between the two groups, and thus is likely related to high serum phosphate [4,5,6,7,8] and high FGF23 concentrations in SCD patients [8,50,51], which would exert an inhibitory effect on calcitriol synthesis by the proximal tubule [49]. Moreover, hypophosphatemia is a frequent feature in pHPT, as PTH inhibits phosphate reabsorption by proximal tubules [52,53,54,55]. In our series, only 2 out of 12 pHPT SCD patients (17%) have hypophosphatemia versus 52 out of 91 pHPT (57%) renal stone patients. We can assume that the tendency for SCD patients to have pre-existing high serum phosphate levels in a basal state explains why pHPT does not yield serum phosphate values within the normal range, despite the excessive excretion of PTH by the parathyroid glands.

Our study may have several limitations: the comparison of biochemical parameters between pHPT SCD and pHPT renal stone patients might include some biases, such as age and ethnicity, as SCD patients are younger and of African or West Indian descent, whereas renal stone patients were mostly Caucasian. Normocalcemic pHPT patients were not included to avoid additional bias, which may imply a higher prevalence of pHPT than indicated in our SCD population. Due to the low number of pHPT SCD and FHH patients in our groups, a comparison was not performed due to a lack of power. However, the biological pattern shown in Figure 1 looks very similar, despite genuine pHPT in SCD patients being responsible for the stress associated with the handling of renal calcium tubular in SCD population. Daily calcium intake and bone status was unfortunately lacking in this study and should be part of the metabolic workup in this population, as the prevalence of very low bone mass density in pHPT SCD patients was reported in one study [21].

To conclude, we confirm a high prevalence of pHPT among our SCD population, with a biological pattern including high serum calcium, inadequate PTH levels and subnormal calcitriol concentrations, and a strikingly low FeCa^2+^, which is associated with chronic hemolysis biomarkers. The precise mechanism at play and the identification of the involved tubular segment remain largely speculative and require further investigation.

## Figures and Tables

**Figure 1 jcm-10-05179-f001:**
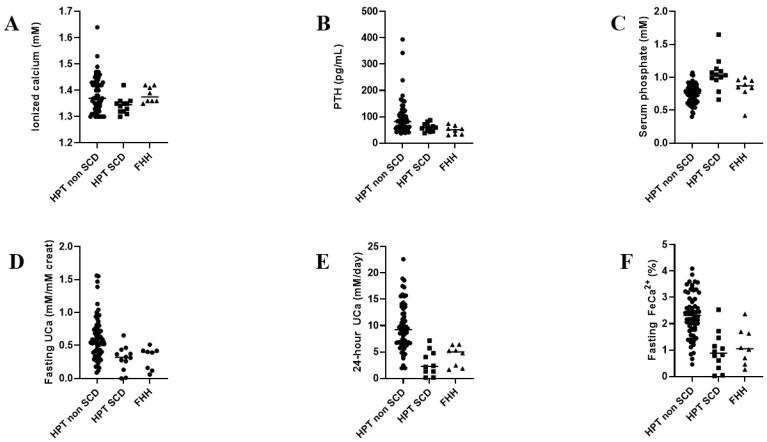
Distribution of different biochemical parameters related to calcium homeostasis among pHPT non-SCD patients, pHPT SCD and FHH patients. (**A**) Ionized calcium (**B**) PTH concentration (**C**) serum phosphate (**D**) fasting urine calcium (expressed as urine calcium/creatinine ratio) (**E**) 24 h urinary calcium excretion and (**F**) fasting FeCa^2+^. Lines indicate median values.

**Table 1 jcm-10-05179-t001:** Comparison of demographic and biological parameters between pHPT non-SCD, pHPT SCD and non-pHPT SCD patients.

	pHPT Non-SCD (*n* = 64)	pHPT SCD (*n* = 12)	Non-pHPT SCD (*n* = 165)	*p*-Value (Kruskal-Wallis Test)
Age (y)	54 (42–61)	43 (34–50)	30 (23–39) ^a,b^	**<0.0001**
Sex (F/M)	46/64	7/12	91/165	0.16
BMI (kg/m^2^)	24.8 (22.2–28.2)	22.0 (20.2–26.5)	21.9 (19.8–24.5) ^a^	**<0.0001**
**Blood**				
eGFR (mL/min/1.73 m^2^)	98 (89–108)	118 (104–132) ^a^	129 (115–144) ^a^	**<0.0001**
Calcium (mM)	2.61 (2.54–2.71)	2.61 (2.49–2.66)	2.33 (2.26–2.41) ^a,b^	**<0.0001**
Ionized calcium (mM)	1.37 (1.32–1.43)	1.35 (1.32–1.36)	1.21 (1.18–1.23) ^a,b^	**<0.0001**
Phosphate (mM)	0.76 (0.63–0.85)	1.04 (0.97–1.12) ^a^	1.21 (1.09–1.37) ^a,b^	**<0.0001**
Magnesium (mM)	0.85 (0.80–0.93)	0.82 (0.73–0.90)	0.82 (0.77–0.88) ^a^	**0.03**
Uric acid (µM)	308 (271–382)	350 (272–525)	377 (301–462) ^a^	**0.0007**
Bicarbonate (mM)	25 (24–27)	24 (23–25)	25 (24–27)	0.17
25-hydroxyvitamin D (ng/mL)	20 (11–27)	14 (10–24)	15 (10–24)	0.22
1,25-dihydroxyvitamin D (pg/mL)	81 (65–102)	52 (38–81) ^a^	49 (39–70) ^a^	**<0.0001**
PTH (pg/mL)	82 (57–113)	59 (45–69)	36 (25–46) ^a,b^	**<0.0001**
FGF23 (RUI/mL)	69 (57–94)	802 (589–1355) ^a^	721 (341–1351) ^a^	**<0.0001**
BALP (ng/mL)	17 (12–25)	19 (15–34)	18 (14–25)	0.30
**Urine**				
DPD (mM/mM creatinine)	8.2 (5.7–10.8)	4.9 (4.2–6.6) ^a^	4.5 (3.2–5.8) ^a^	**<0.0001**
Fasting FeCa^2+^ (%)	2.84 (1.91–3.58)	0.88 (0.40–1.38) ^a^	0.34 (0.13–0.66) ^a^	**<0.0001**
Calcium (mM/day)	9.24 (6.68–12.98)	2.31 (1.04–5.00) ^a^	1.12 (0.55–1.94) ^a^	**<0.0001**
Sodium (mM/day)	127 (86–166)	115 (84–145)	104 (68–136) ^a^	**.02**
Urea (mM/day)	349 (284–425)	188 (157–298) ^a^	255 (175–328) ^a^	**<0.0001**
Creatinine (mM/day)	9.4 (7.6–11.7)	9.5 (5.8–11.5)	10.2 (7.6–13.6)	0.54
24-h urine pH	6.2 (5.8–6.6)	5.5 (5.3–6.3) ^a^	5.7 (5.4–6.0) ^a^	**<0.0001**
NH_4_^+^ (mM/day)	29.0 (21.6–43.6)	17.3 (12.3–26.1) ^a^	19.4 (14.0–25.7) ^a^	**<0.0001**

y, years; F, female; M, male; mM, mmol/L; µM, µmol/L; ^a^: *p* < 0.05 vs. pHPT non-SCD group; ^b^: *p* < 0.05 vs. pHPT SCD group.

**Table 2 jcm-10-05179-t002:** Univariate analysis of biochemical factors associated with fasting FeCa^2+^ in non-pHPT SCD patients, before and after adjustment on PTH. y, years; mM, mmol/L; µM, µmol/L.

	Standardized Coefficient No Adjustment	*p*-Value	Standardized Coefficient Adjustment for PTH	*p*-Value
Age (y)	0.14	**0.01**	0.20	**0.01**
eGFR (mL/min/1.73 m^2^)	−0.02	0.23	−0.10	0.23
Ionized calcium (mM)	0.13	0.11	0.12	0.11
Serum phosphate (mM)	−0.10	0.19	−0.10	0.19
PTH (pg/mL)	−0.26	–	–	-
1,25-dihydroxyvitamin D (pg/mL)	0.06	0.62	0.04	0.62
FGF23 (RUI/mL)	0.05	0.44	0.08	0.44
Hemoglobin (g/dL)	0.21	**0.02**	0.19	**0.02**
Hematocrit (%)	0.23	**0.007**	0.21	**0.007**
LDH (U/L)	−0.25	**0.02**	−0.21	**0.02**
Bilirubin (µM)	−0.16	0.05	−0.17	0.05

## Data Availability

Upon request, data supporting reported results can be obtained from J-P.H. (jean-philippe.haymann@aphp.fr).

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
