# Peer review of "Primary Hyperparathyroidism in Homozygous Sickle Cell Patients: A Hemolysis-Mediated Hypocalciuric Hypercalcemia Phenotype?"

_jcm, 2021, doi:10.3390/jcm10215179_

Round 1
Reviewer 1 Report
The study design is incorrect to answer the scientific question.
The authors are comparing 3 distinct groups of patients retrospectively hoping to find a differences in phenotype. The intent for such a pursuit that drives the methodology is unclear. The lack of controls introduces selection bias (bias due to conditioning on a collider). The differences in the fractional excretion of calcium may be artificial.
Author Response
The aim of our retrospective study is to point out an unexpected (and until now unnoticed) phenotype of primary hyperparathyroidism (pHPT) in homozygous sickle cell patients and furthermore to raise the issue of the underlying pathophysiology.
Indeed, the pathophysiology of pHPT is well-known in renal stone patients, and our results show a phenotype very close to FHH patients suggesting that inactivation of renal CaSR may be at play. We fully acknowledge that this hypothesis, while interesting, remains speculative.
Our study adds further insight into renal calcium handling in non-pHPT SCD patients: indeed, fasting FeCa2+ is very low (0.34%) whereas in a non-SCD normal population, fasting FeCa2+ is usually within 1-2% range (and above 2% in the setting of hypercalcemia). Our finding that fasting FeCa2+ is associated with some hemolytic markers independently of serum PTH is also a new finding and suggests a link between hemolysis and calcium renal tubular handling which should deserve further studies, noteworthy in experimental models, to unravel the mechanisms at play.
Reviewer 2 Report
The evaluated work meets the requirements of a research investigation. It is a well presented and developed paper, whose development hypothesis is confirmed by suitable methods. A very interesting work.
Author Response
We do thank very much the reviewer for his interest and very nice comment indeed.
Reviewer 3 Report
The manuscript deals with an interesting clinical aspect detected in sickle cell disease patients. The manuscript is well set up and also well written. Despite this, some problems are highlighted.
- the description of the FHH patient group is missing. The number of patients is not reported and there are few biochemical data. This is important to have a more complete biochemical picture of the groups in order to better understand the phenotypic similarity between the pHPT-SCD and FHH patients.
- in line 224 the expression "This normal calcitriol value" is incorrect in reference to the pHPT -SCD group, but it should be “This low or subnormal calcitriol value ...”, as it is correctly defined later in the text.
- in line 240 Fig.3 should be changed to Fig.2
- beyond the slight problems noted in the previous points, there are some important limitations in this work. The authors argue that the hypocalciuric hypercalcemia, detected in 6.8% of their patients pHPT-SCD, is promoted by chronic hemolysis. This conclusion, which is also reported in the title of the manuscript, is not exhaustively described in the text but above all not documented and proven with data. The relationship between chronic hemolysis and changes in calcium homeostasis is not understood.
- Among the values shown in Figure 2, the high value of FGF23 in SCD patients compared to non-SCD patients is particularly intriguing. The authors described that the high levels of FGF23 correlate with the low levels of calcitriol that they associate with hyperphosphatemia, found in most their pHPT-SCD patients. However, the role played by FGF23 in the homeostasis of calcium and phosphate is also important, which when it increases inhibits the synthesis of 1,25-dihydroxyvitamin D and the renal reabsorption of calcium and phosphate. The authors rightly speculated that patients with SCD “have in basal state already pre-existing high serum phosphate levels” and that this “explains why pHPT yields serum phosphate values rather within the normal range, despite excessive excretion of PTH by the parathyroid glands”, but in this hypothesis they did not consider the role of FGF23. It would be important to also evaluate the functionality of FGF23 and the responsiveness of the kidneys to FGF23. For this purpose it would be advisable to exclude mutations or polymorphisms of FGF23 or especially its receptor complex Klotho/FGFR, in relation to the fact that pHPT-SCD patients are negative for CASR mutations. The same authors honestly state that there are several limitations in the study, thus even if the study has been correctly set up, perhaps the results obtained are yet partial and preliminary in order to clearly identify and define the hypocalciuric hypercalcemia phenotype in SCD patients likely hemolysis-mediated.
Author Response
Reviewer n°3
The manuscript deals with an interesting clinical aspect detected in sickle cell disease patients. The manuscript is well set up and also well written. Despite this, some problems are highlighted.
the description of the FHH patient group is missing. The number of patients is not reported and there are few biochemical data. This is important to have a more complete biochemical picture of the groups in order to better understand the phenotypic similarity between the pHPT-SCD and FHH patients.
Thank you for your suggestions. We modified the result section as followed:
Eight patients (2 males and 6 females) with a final diagnosis of FHH confirmed by genetic sequencing are reported in the present study, with a median age of 41 years old (IQR: 31-47). Median ionized calcium and PTH were 1,38 mmol/L (IQR: 1,36-1,42) and 51 pg/mL (IQR: 33-64) respectively.
As shown on Figure 1, pHPT SCD patients biochemical pattern appears indeed very similar to FHH patients, in particular 24-hour urine calcium (median value at 3,42 mmol/day, IQR: 2,21-5,15) and fasting FeCa2+ (median value at 1,05%, IQR: 0,52-1,68).
in line 224 the expression "This normal calcitriol value" is incorrect in reference to the pHPT -SCD group, but it should be “This low or subnormal calcitriol value ...”, as it is correctly defined later in the text.
Thank you for your notice. The modification was performed in this new version.
in line 240 Fig.3 should be changed to Fig.2
Thank you for your notice. The modification was performed, but we actually changed it into Fig. 1.
beyond the slight problems noted in the previous points, there are some important limitations in this work. The authors argue that the hypocalciuric hypercalcemia, detected in 6.8% of their patients pHPT-SCD, is promoted by chronic hemolysis. This conclusion, which is also reported in the title of the manuscript, is not exhaustively described in the text but above all not documented and proven with data. The relationship between chronic hemolysis and changes in calcium homeostasis is not understood.
We acknowledge that the relationship between chronic hemolysis and changes in calcium homeostasis in SCD patients is at present not understood. Nonetheless, we state the following in SCD patients who undergo a chronic hemolysis since birth: 1) fasting FeCa2+ is surprisingly low in pHPT patients ; 2) in non-pHPT SCD patients, FeCa2+ is also low (in general population, FeCa2+ is expected between 1-2% range and above 2% in the setting of hypercalcemia) and of notice, significantly associated with the hemolytic biomarker LDH, independently of serum PTH. Thus, as discussed, enhanced tubular calcium handling in SCD patients is very efficient and likely promoted by chronic hemolysis (rather than vaso-occlusive phenomena), though the causal link is missing. Long term kidney injury is reasonably ruled out (no link with albuminuria or eGFR is found, but also urinary ammonium excretion (P = 0,20) and urine osmolality (P = 0,30)).
The potential role of lower calcitriol synthesis, higher serum FGF23, higher serum phosphate (and/or klotho) could not be documented and thus the mechanisms at play remain to be unraveled. We speculate that a missing metabolite such as HIF1, heme or heme oxygenase, mediated by chronic anemia and hemolysis may be involved in the process. However, no data in the literature support this view at present and we feel uncomfortable to speculate further in the discussion section...
We modified the discussion as followed : Though chronic hemolysis in SCD patients is associated with a systemic macro and microangiopathy, leading to organ damages such as pulmonary hypertension, leg ulcers or priapism [43, 44], no link was found with renal function or albuminuria which were reported to be associated with chronic hemolysis [41, 42]. Moreover, renal medulla and TALH lesions which are considered a consequence of vaso-occlusive events are an unlikely hypothesis to explain a partial loss of CaSR activity resulting in enhanced tubular calcium reabsorption. Indeed, no association was found between fasting FeCa2+ and fasting urine osmolality or urinary ammonium excretion (data not shown).
Among the values shown in Figure 2, the high value of FGF23 in SCD patients compared to non-SCD patients is particularly intriguing. The authors described that the high levels of FGF23 correlate with the low levels of calcitriol that they associate with hyperphosphatemia, found in most their pHPT-SCD patients. However, the role played by FGF23 in the homeostasis of calcium and phosphate is also important, which when it increases inhibits the synthesis of 1,25-dihydroxyvitamin D and the renal reabsorption of calcium and phosphate. The authors rightly speculated that patients with SCD “have in basal state already pre-existing high serum phosphate levels” and that this “explains why pHPT yields serum phosphate values rather within the normal range, despite excessive excretion of PTH by the parathyroid glands”, but in this hypothesis they did not consider the role of FGF23. It would be important to also evaluate the functionality of FGF23 and the responsiveness of the kidneys to FGF23. For this purpose it would be advisable to exclude mutations or polymorphisms of FGF23 or especially its receptor complex Klotho/FGFR, in relation to the fact that pHPT-SCD patients are negative for CASR mutations. The same authors honestly state that there are several limitations in the study, thus even if the study has been correctly set up, perhaps the results obtained are yet partial and preliminary in order to clearly identify and define the hypocalciuric hypercalcemia phenotype in SCD patients likely hemolysis-mediated.
We do thank the reviewer for his relevant comments and raising this interesting issue. Indeed, whereas high serum FGF23 concentrations could account for a decreased calcitriol level in pHPT SCD patients compared to pHPT non-SCD patients, decreased calcitriol is expected, given the same ionized calcium and PTH values, to increase FeCa2+ through its action on distal renal tubules which is not the case in our population.
However, as you suggested, FGF23 may promote increased calcium reabsorption in the distal tubule through an enhanced TRPV5 expression [39]. Alternatively, increased soluble Klotho is also a very exciting hypothesis to explain increased calcium tubular reabsorption in the distal tubular segment acting presumably on TRPV5. Indeed, secreted Klotho protein inhibits TRPV5 internalization through its sialidase activity on this ion channel, increasing TRPV5 cell surface levels and Ca2+ tubular reabsorption (37, 38). However, increased soluble Klotho concentration would not explain high serum phosphate levels and thus could not explain the whole picture (to our knowledge, no data are available on soluble klotho concentration in SCD patients).
We modified the discussion section as followed: “Nevertheless, an increased soluble klotho concentration, yet to be confirmed, could account for TRPV5 activity increase in this tubular segment independently of PTH and calcitriol concentrations [37,38]. Alternatively, enhanced TRPV5 expression could also be due to FGF23 activity [39]. Conversely, the thick ascending loop of Henle (TALH) and CaSR could also be potential keyplayers.”
Additional references :
- Chang Q, Hoefs S, van der kemp AW et al. The beta-glucoronidase klotho hydrolyzes and activates the TRPV5 channel. Science. 2005 Oct 21;310(5747):490-3.
- Cha SK, Ortega B, Kurosu H et al. Removal of sialic acid involving Klotho causes cell-surface retention of TRPV5 channel via binding to galectin-1. Proc Natl Acad Sci U S A. 2008 Jul 15;105(28):9805-10.
- Andrukhova O, Smorodchenko A, Egerbacher M et al. FGF23 promotes renal calcium reabsorption through the TRPV5 channel. EMBO J. 2014 Feb 3;33(3):229-46.

Round 2
Reviewer 3 Report
The authors have made corrections and taken note of the issues raised by me. The discussion seems more complete and more suggestive of the likely relationship between chronic hemolysis and the change in calcium homeostasis.